# Neuroprotective Effects of Blueberries through Inhibition on Cholinesterase, Tyrosinase, Cyclooxygenase-2, and Amyloidogenesis

Pari Samani, Sophia Costa and Shuowei Cai *

Department of Chemistry and Biochemistry, University of Massachusetts Dartmouth,
North Dartmouth, MA 02747, USA
* Correspondence: scai@umassd.edu; Tel.: +1-508-999-8807

**Abstract:** Blueberries are rich in polyphenolic compounds and have shown improvement in cognitive function in several clinical trials. The molecular basis of the neuronal protection of blueberries, however, is not fully understood. The objective of this research is to understand the biochemistry basis of neuronal protection effects of blueberries through their impacts on several enzymes and pathways involved in Alzheimer's disease (AD) and other neurodegenerative diseases. We examined the inhibition effects of blueberries on the enzymatic activity of cholinesterase (acetylcholinesterase, AChE; and butyrylcholinesterase, BuChE), tyrosinase, and cyclooxygenase-2 (COX-2). The effects of blueberries on the biosynthesis of acetylcholinesterase in a cellular model were also studied. Further, the effect of blueberries on amyloid fibril formation was evaluated. Our results showed that blueberries directly inhibit the enzymatic activity of AChE, BuChE, tyrosinase, and COX-2, with the IC$_{50}$ at 48 mg/mL, 9 mg/mL, 403 mg/mL, and 12 mg/mL of fresh berry equivalent, respectively. Further, blueberries delay the amyloid fibril formation by 24 h at 39 mg fresh berry/mL. It also reduces the synthesis of acetylcholinesterase synthesis at 19 mg fresh berry/mL in a cellular model. Those results suggested that the neuroprotection effects of blueberries may involve different pathways, including enhancing cholinergic signaling through their effect on cholinesterase, reducing neuroinflammation through inhibition of COX-2, and reducing amyloid formation. Collectively, blueberries may play a vital role in neuronal protection beyond their antioxidant activity and our results provide more molecular mechanisms for their neuroprotective effects, and support blueberries being nutraceutical to improve cognitive function.

**Keywords:** blueberry; neuroprotection; Alzheimer's disease; cholinesterase; acetylcholinesterase; butyrylcholinesterase; tyrosinase; cyclooxygenase 2; amyloid; neurodegeneration





## 1. Introduction

Blueberries (*Vaccinium* genus) are rich in polyphenolic compounds, especially flavonoids and anthocyanins. Polyphenol-rich foods have been reported to reduce the risk of age-related diseases that constitute major socioeconomic burdens, including neurological decline, cardiovascular disease, and type 2 diabetes mellitus [1,2]. Blueberries have been of interest particularly due to their high contents of antioxidants and polyphenols. In several observational clinical studies, blueberries have shown their neuroprotection. In a prospective analysis of 16,000 women in the landmark Nurses' Health Study, a greater intake of blueberries has been shown to associate with slowing cognition decline in older women by an estimate of 2.5 years [3]. Small placebo-controlled trials studying the effect of blueberries on cognitive function showed that blueberries improve cognition among healthy older adults and midlife adults [4,5]. In a pooled analysis of two US cohort studies that examined almost 150,000 people, lower Parkinson's disease risk was associated with the highest quintile of anthocyanin and berry intake [6].

Dementia is the loss of cognitive function. Alzheimer's disease (AD), the most common form of dementia, is a neurodegenerative disease affecting the structural integrity of the brain and accounts for 60 to 80 percent of dementia cases [7]. While the causes of AD are still unknown, several hypotheses have been proposed: deficits in the cholinergic transmission, beta-amyloid plagues (Aβ), tau tangles, oxidative damage and mitochondrial dysfunction, neuronal inflammation, synapse loss, vascular changes; endosomal abnormalities, among others [8,9]. Genetics also plays a role in AD, such as the mutation of genes for the apolipoprotein E epsilon 4 allele (APOE-ε4), the amyloid precursor protein (APP), and presenilin (PS1 & PS2) [10]. Several of those hypotheses are interlinked, and collectively, they lead to neuron death. For example, cholinesterase (including acetylcholinesterase and butyrylcholinesterase) is a critical enzyme to regulate the level of the neurotransmitter and thus plays important role in cholinergic signaling. Both Aβ and abnormally hyperphosphorylated tau can increase acetylcholinesterase (AChE) expression [11]. The increased AChE further influences PS1 and tau-protein kinase GSK-3β. GSk-3β induces hyperphosphorylated tau (P-tau), while PS1 affects APP processing and Aβ production [11,12]. Aβ polymerizes to form plaques that deposit in the brain disrupting neuronal cell function. This cascade induces neuroinflammation and oxidative stress, which leads to cognitive decline [13]. Like AChE, butyrylcholinesterase (BuChE) hydrolyzes the neurotransmitter acetylcholine (butyrylcholine, the preferred substrate for BuChE, is not existent naturally), but at a much slower rate. While it is less abundant in the brain and considered to play a minor role in regulating the acetylcholine level in the brain, BuChE is found to compensate for AChE when its levels deplete [14]. BuChE activity has been found to progressively increase in AD patients [15]. Further, BuChE is also found to involve the Aβ plaques maturation [16]. Therefore, both AChE and BuChE are valid targets for AD [14,17]. In addition to AD, both AChE and BuChE are also found to play roles in multiple sclerosis [18], and higher levels of AChE and BuChE are observed in patients with Parkinson's disease dementia compared with Parkinson's disease [14].

Neuronal inflammation is another key factor leading to the damage of neuronal cells. Accumulation of aggregated proteins and damaged neurons causes inflammation as per reports, along with imbalances between pro- and anti-inflammatory processes [19]. Pathogenic stimuli, such as the accumulation of abnormal proteins in cells or extracellular spaces (including Aβ), cause cellular stress responses resulting in the progressive dysfunction and deration of neurons [20]. Cyclooxygenase-2 (COX-2) is involved in the biosynthesis of prostaglandins (PGs) under severe inflammation. Elevated PGs are reported to be involved in the pathogenesis of AD [21]. Tyrosinase oxidizes dopamine to form melanin pigments through the formation of dopamine quinone, a reaction that results in the formation of highly reactive oxygen or nitrogen species capable of inducing neuronal cell death [22]. Oxidation stress links to both inflammation and endosomal abnormalities, which may cause damage to neurons and has been hypothesized as potential links to Parkinson's disease [22,23].

AD is not caused by any single factor, and the above-mentioned hypotheses are interconnected. Therefore, the protection of the neurons from damage by different factors is the key to preserving the normal functions of neurons. This is difficult to be accomplished through a single drug or even multiple drugs. Lifestyle changes, however, may hold keys to the prevention of AD, and potentially even decelerate the progression of the disease. Consumption of phenolic compounds-rich foods like blueberry is part of a healthy lifestyle that may contribute to neuroprotection and help slow the decline of cognition function. The detailed mechanisms of the neuroprotection effects of blueberries are still unknown. Most current studies on the health benefits of blueberries, including their neuroprotection, are focused on their antioxidant activities [24–27]. Recently, it is also found that blueberry inhibits monoamine oxidase A, an enzyme that induces oxidative stress [28]. While there is a report that blueberry inhibits AChE and BuChE, no systematic studies have been done to identify how effectively blueberries inhibit them. Further, no studies have been done on the effects of blueberries on COX-2, and amyloidogenesis. This study aimed

to quantify the main classes of the polyphenols of blueberry extract and understand the role of blueberries in the protection of neuronal cells, by examining their effects on key enzymes involving neuronal degeneration: cholinesterase (AChE and BuChE), tyrosinase, and COX-2, along with the amyloid fibril formation. Those will provide the biochemical basis of the neuroprotection of blueberries.

## 2. Materials and Methods

### 2.1. Chemicals and Materials

Blueberries were purchased from a local supermarket in Dartmouth, Massachusetts, USA. COX-2 (human) Inhibitor Screening Assay Kit was purchased from Cayman Chemicals (Ann Arbor, MI, USA). The following were purchased from Sigma Aldrich (St. Louis, MO, USA): Acetylcholinesterase (AChE) from the electric eel, Tyrosinase from mushroom, galantamine, acetylthiocholine iodide, kojic acid, L-DOPA, Hen's egg white lysozyme and MTT assay kit. Dulbecco's Modified Eagle's Medium (DMEM-for cell culture growth) sterile containing 2 mM glutamine, fetal bovine serum (FBS), trypsin-EDTA (0.05%), DTNB (5,5-dithio-bis-2-nitrobenzoic acid), 3,3′,5,5′-Tetramethylbenzidine (TMB), trypan blue solution 0.4%, INCYTO DHC-N01 disposable hemocytometer, corning® cell culture flasks having surface area 25 cm$^2$, Corning® Costar® TC-Treated 6 Well Plates (Poly-D-Lysine coated), and 96 well flat bottom non-sterile plates were purchased from Thermo Fisher Scientific (Waltham, MA, USA). Mouse monoclonal antibody IgG2a raised against amino acids 481-614 of AChE of human origin (SC-373901) and Goat anti-Mouse IgG-HRP (horseradish peroxidase, SC-2005) were purchased from Santacruz Biotechnology (Dallas, TX, USA). Congo Red was purchased from MP Biomedicals and 99% pure Nile Red from ACROS Organics.

SpectraMax M5 Multi-Mode Microplate reader (Molecular Devices, San Jose, CA, USA) with SoftMax Pro software was used for microplate-based assays. Absorbance spectra were collected using Shimadzu UV-2450 UV-Vis Spectrophotometer with the UVProbe software. Fluorescence and turbidity were measured using the FluoroMax Spectrofluorometer with the Fluor Essence software (Horiba Scientific). Cell morphology was monitored using Olympus Microscope (Olympus Corp., Tokyo, Japan) and the pictures of cell morphology were taken using an inverted fluorescence microscope, DMI8 (Leica Microsystems, Wetzlar, Germany).

### 2.2. Methods

#### 2.2.1. Extraction of Blueberries

500 g of blueberries were washed and stored at −20 °C for 30 min before the extraction process. The extraction solvent used for the polar compounds was a mixture of acetone, methanol, water, and formic acid (40:40:19:1 *v/v/v/v*). Polar compounds from blueberries were extracted by blending blueberries with the solvent mixture and vacuum filtered. The solvent from the filtrate was eventually evaporated using a vacuum rotary evaporator (Rotavapor II SJ24/40, A, 100-120V from Buchi) at 28 °C to obtain the crude viscous blueberry extract. The resulting extract was then de-sugared and eluted using methanol on a Diaion resin (Sigma Aldrich, St. Louis, MO, USA). Methanol was rotary evaporated and the aqueous blueberry solution was lyophilized using FreeZone 4.5 Liter Benchtop Freeze Dry System (Kansas City, MO, USA). The solid extract obtained contains polar compounds from blueberry (referred to as blueberry extract).

#### 2.2.2. Quantification Assays Using Colorimetric Methods

- Quantification of total phenolic content

The total phenolic content of the blueberry extract was determined using the Folin–Ciocalteu method [29]. The diluted aqueous blueberry extract (500 μL) was mixed with Folin–Ciocalteu reagent (2.5 mL, 0.2N) and incubated for 5 min at room temperature followed by the addition of 2 mL of sodium carbonate solution (75 g/L). The mixture was incubated for 1 hr at room temperature and the absorbance was read at 765 nm against

a water blank. A standard calibration curve was plotted using gallic acid (0–150 mg/L). Results were expressed as gallic acid equivalents.

- Quantification of total flavonoids content

The total flavonoids in the blueberry extract were determined based on the formation of the flavonoid-aluminum complex [30]. Briefly, to 1000 µL of the test solution (standard or diluted blueberry extract sample in methanol), 300 µL of sodium nitrite (5% *w/v*) was added. The solution was rested for 5 min at room temperature and 500 µL of $AlCl_3$ (2% *w/v*) was added. The mixture was incubated at room temperature for 6 min and neutralized with 500 µL sodium hydroxide solution (5M). The mixture was incubated for 10 min at room temperature and spectral analysis was carried out in the range of 350–600 nm using a 1 cm cuvette with Shimadzu UV-2450 UV-Vis Spectrophotometer. The $AlCl_3$ solution was substituted by an equal volume of methanol in the blank. Rutin was used as a flavonoid standard with a concentration range of 0–350 mg/L. The absorbance was measured at 510 nm for quantitative analysis of flavonoid content. Results were expressed as rutin equivalents.

- Quantification of total anthocyanins content

A pH differential method was employed using two buffers- hydrochloric acid-potassium chloride (pH 1, 0.2 M) and acetic acid-sodium acetate (pH 4.5, 1 M) using a 1 cm cuvette [31]. 100 µL of the diluted aqueous blueberry extract was mixed with 900 µL of corresponding buffers. The sample was incubated for 15 min, and absorbance was measured at 510 and 700 nm using a 1 cm cuvette. Absorbance was calculated as $A = [(A_{510}-A_{700})\, pH_1]-[(A_{510}-A_{700})\, pH_{4.5}]$, and the concentration of total anthocyanins was determined using the extinction coefficient of 26,900 $M^{-1}cm^{-1}$, based on the cyanidin-3-glucoside.

### 2.2.3. In Vitro Enzymatic Inhibition

The effect of blueberries on key enzymes was evaluated in vitro and its inhibitory impact was calculated as follows

$$\% \text{ Inhibition} = \frac{E - S}{E} \times 100 \qquad (1)$$

where E is the activity of the enzyme without the inhibitor (negative control) and S is the activity of the enzyme with the inhibitor. $IC_{50}$ of enzymatic inhibition was defined as absolute inhibition of 50% of the enzymatic activity of respective enzymes, using the non-linear regression of the dose–response curve [32].

- Cholinesterase inhibition

AChE and BuChE activity was measured using the spectrophotometric method developed by Ellman et al. in a microplate format to evaluate the inhibition effect of blueberry extract [33]. Galantamine, a selective inhibitor of AChE was used as a positive control (0.01 mg/mL). Four concentrations of blueberry extract in 50 mM tris buffer (pH = 7.4) were used to determine the dose-response on the inhibition of AChE and BuChE. The substrate for AChE and BuChE is acetylthiocholine iodide and butyrylthiocholine chloride, respectively. ChE-catalyzed hydrolysis of the thiocholine results in the formation of thiocholine which is indicated by the formation of the yellow 5-thio-2-nitrobenzoate anion as a result of the reaction of DTNB with thiocholines. The reaction rate was attained by reading the absorbance at 405 nm for 10 min at the interval of 1 min using the SpectraMax M5 microplate reader. The percentage of AChE/BuChE inhibition was determined using Equation (1).

- Tyrosinase inhibition

The effect of blueberry extract on the activity of tyrosinase was spectrophotometrically measured using L-DOPA (6.28 mM) as the substrate in 96-well microplate format as reported previously with modifications [34]. Kojic acid (0.27 mg/mL, stock) a known inhibitor

of tyrosinase was used as a positive control. Tyrosinase from mushroom (0.003 mg/mL) was used. Phosphate buffer (50 mM, pH = 6.5) was used for the dilution of reagents and blueberry extract samples. Phosphate buffer was added followed by the enzyme (15 μL) and inhibitor (35 μL) in a 96-well plate. The plate was incubated at room temperature for 10 min and the substrate (55 μL) was added. The plate was incubated for 30 min at room temperature. Absorbance was read at 490 nm as an endpoint using the Spectramax M5 microplate reader and the percentage inhibition was calculated using Equation (1).

- Cyclooxygenase-2 (COX-2) inhibition

Inhibition of COX-2 by the blueberry extract was assessed by monitoring the quantity of prostaglandins using a competitive enzyme-linked immunosorbent assay (ELISA). COX-2 (human) Inhibitor Screening Assay Kit 701080 (Cayman Chemical, Ann Arbor, MI, USA) was used for the assay with celecoxib, a known COX-2 selective inhibitor as the positive control. Cayman's protocol was followed. PG generated from the enzymatic reaction was quantitated using competitive ELISA with rabbit antiserum against PG. The percentage of COX-2 inhibition was calculated based on Equation (1).

### 2.2.4. Effect of Blueberry Extract on AChE Synthesis

Human neuroblastoma cell line M17 was used to examine the effects of blueberry extract on the synthesis of AChE. The AChE produced from the cells was measured using indirect ELISA with the monoclonal antibody against human AChE. The cells were seeded in poly-D-lysine coated 6-well plate at a density of $1.5 \times 10^5$ cells/well and maintained with DMEM media containing 10% FBS in the $CO_2$ incubator at 37 °C having 95% humidity and 5% $CO_2$. After 24 h, cells were treated with different concentrations of blueberry extract (0 to 0.5 mg/mL) in serum-free media for 48 h. Post-treatment, the morphology of cells was examined using an Olympus CK 2 microscope (Olympus Corp., Tokyo, Japan), and the resulting number of cells was counted using a hemacytometer. The viability of cells was assessed using the MTT assay. The cells were thoroughly washed with PBS and harvested into a pre-weighed Eppendorf tube. The cells were then pelleted by centrifugation, and the supernatant was carefully removed. The Eppendorf tubes were re-weighed to determine the weight of the pellet. Further 50 μL of lysis buffer was added to each tube and sonicated to extract the protein from cells. 300 μL of cold PBS was then added to each tube and mixed well. The concentration of protein in the cell lysate was determined using the Bradford assay. The quantity of AChE in cell lysate was determined using an indirect ELISA method. Briefly, the cell lysate samples were diluted to normalize the protein concentration and 100 μL of the sample was coated on a Nunc MaxiSorp 96-well microplate. The parafilm-covered plate was incubated overnight at 4 °C to allow the binding of the protein on the plate surface. The solution was decanted and washed with PBST (PBS containing 0.05% tween). The assay was performed in the order of, blocking using 5% milk, addition of primary monoclonal antibody from the mouse in the ratio of 1:300, and addition of secondary antibody-HRP labeled goat anti-mouse in the ratio 1:2000. A 100 μL of each reagent was added, incubated at 37 °C for one hour and the solutions were decanted post-incubation. The plate was washed using PBST between subsequent steps. A 100 μL of TMB substrate was added and the plate was incubated at room temperature for 15–30 min until the blue color was observed. The reaction was stopped using 50 μL of 2 M sulfuric acid. The plate was read at 450 nm using the Spectramax M5 microplate reader. The reduction in synthesis of AChE was calculated as follows:

$$\%\text{AChE reduction} = \frac{\text{Absorbance of untreated cells} - \text{Absorbance of treated cells}}{\text{Absorbance of untreated cells}} \qquad (2)$$

The statistical analysis was carried out to compare the effects of blueberry extract on AChE synthesis using IBM SPSS Statistics 28 (IBM Corp., Armonk, NY, USA).

2.2.5. Amyloid Fibril Assays

- Amyloid Fibril Formation

Hen egg white lysozyme (HEWL) aggregation was induced using an acidic pH environment and agitation under elevated temperature [35]. Lysozyme solution at 1 mg/mL (70 µM) was prepared in a 50 mM glycine buffer (pH 2.0) and agitation was induced by stirring using a magnetic stir bar spun at 170 rpm at 70 °C for eight days. The aggregation was monitored both by a visual examination as well as the absorbance measurement.

- Congo Red (CR) binding assay

The binding of CR to amyloids induces the characteristic increase in CR absorption leading to a red shift and the presence of a unique shoulder peak at approximately 536 nm. The aggregation was induced as mentioned above. To visualize aggregation via CR binding assay, a 1:1 ratio of 70 µM lysozyme solution in 50mM glycine buffer, pH 2.0, and 20 µM CR in $H_2O$ were mixed and incubated at room temperature for 30 min. The absorbance spectrum was scanned using Shimadzu UV-2450 UV-Vis Spectrophotometer from 450–600 nm post incubation with buffer alone as the blank. The absorbance was measured at 509 nm and 536 nm to monitor the shift due to the amyloid formation. The solution was subjected to aggregation and heated at the same condition for 8 days and the spectrum/absorbance was recorded every 24 h starting from Day 0, the day the solutions were created. The 50 mM glycine buffer was used as a blank. HEWL solution was used as a control. For blueberry samples, HEWL was prepared at 70 µM in 0.2 mg/mL of blueberry extract in 50 mM glycine and the same procedure was followed to study the effect of blueberries on aggregation and CR binding. The 0.2 mg/mL of blueberry extract in 50 mM glycine was used as a blank and the spectrum/absorbance was recorded. The ratio of absorption at 509 nm and 536 nm was recorded for each trial.

- Nile Red (NR) binding assay

NR was used to probe the hydrophobic surface during amyloid fibril formation. NR stock solution (2.4 mM) was prepared in pure ethanol and the working solutions were prepared by diluting the stock solution in the glycine buffer (pH 2.0). To measure the fluorescence of NR binding to amyloid fibrils, a 1:1 ratio of 10 µM HEWL solution to 10 µM NR in glycine buffer was mixed, and sample fluorescence emission spectrum was recorded from 550–800 nm with the excitation wavelength at 530 nm for 8 days every 24 h from day 0. Samples were prepared in the same way as the CR assay with agitation at 170 rpm and 70 °C. The blueberry-treated HEWL samples were prepared in a similar way by making a 70 µM lysozyme sample in 0.2 mg/mL blueberry extract. The wavelength shift, turbidity, and fluorescence were assessed for HEWL alone, and blueberry-treated HEWL samples. The sample measurements were corrected using the respective blank.

- Turbidity measurement through light scattering

The sample turbidity was examined to visualize the effect of blueberries on the formation of amyloid fibrils. Like NR fluorescence, lysozyme control samples and blueberries treated samples were aged at 70 °C and 170 rpm on a stir plate. The samples were read daily from day 0 (fresh sample) through day 8. The light scattering was monitored using FluoroMax Spectrofluorometer at the excitation wavelength and the emission wavelength of 800 nm. The reading from the 90-degree angle on the spectrofluorometer further reduced the interferences from sample absorbance. The sample measurements were corrected using the respective blank used.

## 3. Results and Discussion

### 3.1. Extraction

In this study, the extraction was carried out using a solvent mixture of varying polarities. No previous study has been reported to perform the extraction of blueberries using the current method to the best of our knowledge. The extraction of polar compounds from 500 g of fresh blueberries yielded 778 mg of the extract. Water counts more than 84%

mass of blueberries, followed by carbohydrates (14.5%), protein (0.7%), and fat (0.3%), thus only less than 0.5% mass of fresh blueberries is counted as micronutrients [36]. Therefore, the solvent mixture we used (acetone, methanol, water, and formic acid with 40:40:19:1 *v/v/v/v*) efficiently extracted the polar compounds from fresh berries. The combined vacuum rotary evaporation and lyophilization removed all organic solvents and water during the extraction. However, trace residual solvents may exist in the dried blueberry extract, and organic solvents like methanol and acetone are toxic. We have exploited using non-toxic solvent ethanol for the extraction of blueberry polar compounds and showed that ethanol can efficiently extract the bioactive polar compounds from blueberries (data to be published soon).

### 3.2. Total Phenolics, Flavonoid, and Anthocyanin Content

Berries, especially blueberries, are rich in phenolic compounds, including flavonoids and anthocyanins. The spectrophotometric method provides an easy and convenient way to determine the contents of total phenols, flavonoids, and anthocyanins, and has been widely used [37]. However, relative standards must be used for the quantitation, and the response from individual compounds may be different from the relative standards used. The result from this study indicates that the main polar portion of blueberries is phenolic compounds, especially flavonoids.

The Folin–Ciocalteu reagent was used to determine the content of total phenolic compounds with gallic acid as the relative standard; the aluminum complex formation method for total flavonoids with rutin as the relative standard; and different pH colorimetric method for total anthocyanins with cyanidin-3-glucoside as standard. Most of the polar extracts are flavonoids and phenolic compounds, including anthocyanins. The total phenolic compound content of the blueberry extract is listed in Table 1. More than 50% of the extract belongs to phenolic compounds (559 μg/mg extract, in terms of gallic acid). About one-third of the total phenolic content is anthocyanins (155 μg/mg extract, in terms of cyanidin-3-glucoside). The total flavonoid content (in terms of rutin, 894 μg/mg extract) is more than the total phenolic compounds, which could be due to the different reference standards used. Based on the extraction yield of the blueberry extraction of 1.56 mg extract/g fresh berry (778 mg extract from 500 g fresh berries), the total flavonoids in fresh berries are 1.39 mg/g fresh berries (Table 1).

**Table 1.** Total phenolics, flavonoids, and anthocyanins content in blueberry.

| Compound Class | μg/mg Blueberry Extract * | mg/g Fresh Blueberries † |
|---|---|---|
| Phenolic content (gallic acid equivalent) | 559 ± 22 (*n* = 15) | 0.87 |
| Flavonoid content (rutin equivalent) | 894 ± 88 (*n* = 8) | 1.39 |
| Anthocyanin content (cyanidin-3-glucoside equivalent) | 155 ± 13 (*n* = 5) | 0.24 |

* The contents are shown in Table as the average ± standard deviation, and the number in the parenthesis is the replicate numbers of each assay used for the calculation of the average and standard deviation. † The fresh berry equivalent was calculated based on the extraction yield of 778 mg/500 fresh berries.

### 3.3. In Vitro Enzymatic Inhibition

- Acetylcholinesterase Inhibition Activity:

As shown in Figure 1, the blueberry extract showed strong inhibition of both AChE and BuChE activity in a dose-dependent manner. Galantamine, an FDA-approved selective AChE inhibitor, was used as a positive control for our assay, showing 93% (±1%) and 59% (±6%) inhibition at 0.01 mg/mL for AChE and BuChE, respectively. The $IC_{50}$ of the blueberry extract was calculated using the non-linear regression of the dose-dependent inhibition curve and is estimated as 75 μg/mL, which is equivalent to 48 mg fresh berry/mL

for AChE, and 14 μg/mL (9 mg/mL fresh berry equivalent) for BuChE. AChE inhibitors (galantamine, donepezil, and rivastigmine) are FDA-approved drugs for AD [38]. As a comparison, it has been reported the IC$_{50}$ of galantamine (an FDA-approved drug for AD) is 0.35 μM (0.13 μg/mL) using erythrocyte AChE [39], and 1.45 μg/mL when using AChE from electric eel [40] (AChE from electric eel was used in this study). While weaker than galantamine, blueberries can be consumed in much larger quantities as fruits. The intake of blueberries, therefore, may help manage the symptoms of AD via the AChE inhibitory effect. Importantly, there is some clinical evidence showing that the dual AChE and BuChE inhibitor, rivastigmine, has some better efficacy than selective AChE inhibitor donepezil, and provides the rationale for dual inhibition of BuChE and AChE in the management of AD [14]. Our results here showed that blueberry effectively inhibits both BuChE and AChE at μg/mL (low mg/mL fresh berry equivalent), suggesting long-term consumption of blueberry could help increase the cholinergic signaling and improve the cognition function. Earlier research has shown the inhibition effect of blueberries on the enzymatic activity of AChE and BuChE [27]. We are the first group to quantitatively characterize the dose-dependence of inhibition of AChE and BuChE by blueberries and identified the IC$_{50}$ of blueberry extract on AChE and BuChE. The dual inhibition of AChE/BChE by blueberries at low milligram/mL fresh fruit equivalent may contribute to their neuronal protection and provide benefits for cognition functions.

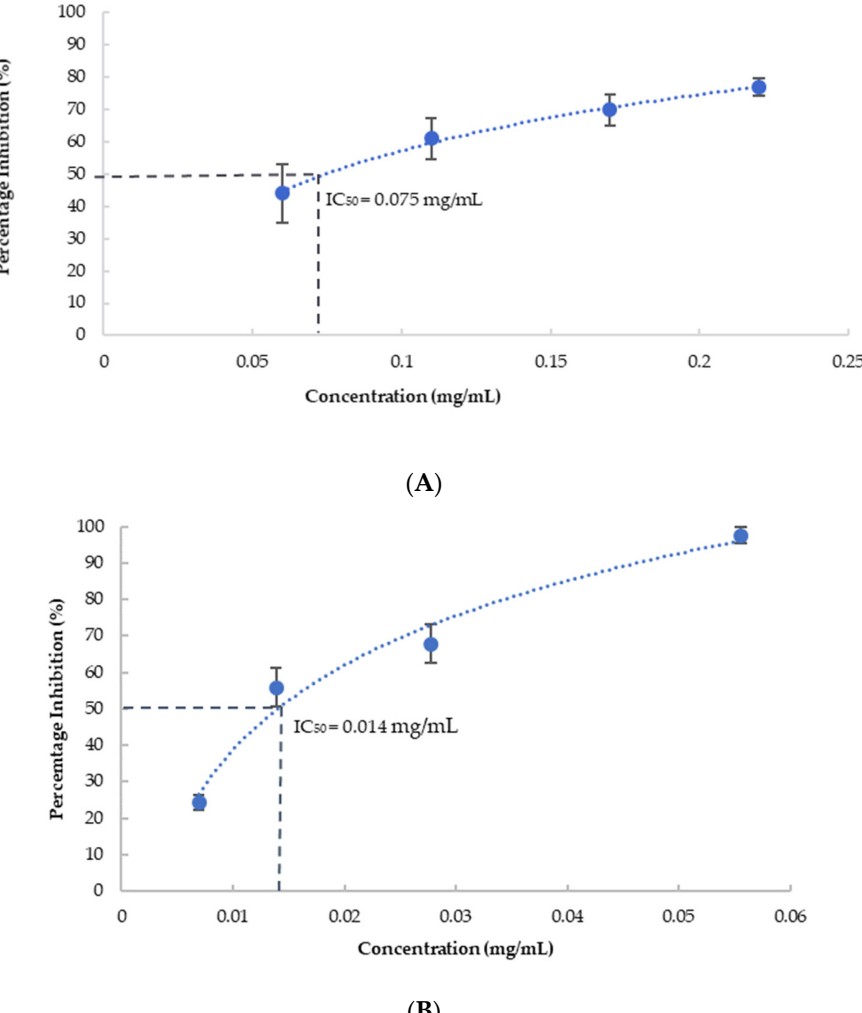

**(A)**

**(B)**

**Figure 1.** Percentage inhibition of cholinesterase by blueberry extract at various concentrations. (**A**): The inhibition of AChE; (**B**): The inhibition of BuChE. Values are presented as the mean ± Standard Deviation (SD) of *n* = 6.

- Tyrosinase inhibition activity:

The blueberry extract substantially inhibited tyrosinase activity in a dose-dependent fashion as shown in Figure 2. $IC_{50}$ was estimated at 627 μg/mL, equivalent to 403 mg fresh berry/mL. Kojic acid, a known tyrosinase inhibitor, was used as the positive control and showed 94% inhibition of tyrosinase at 0.27 mg/mL. Tyrosinase, a copper-containing enzyme, plays a critical role in the synthesis of melanin and results in the formation of reactive oxygen species resulting in cell death. Age-dependent neuromelanin accumulation plays an important role in Parkinson's disease and other neurodegenerative diseases [41]. Neuromelanin is also associated with alpha-synuclein aggregation leading to neurodegenerative diseases [42]. Tyrosinase has been proposed to contribute to the formation of neuromelanin which is associated with alpha-synuclein aggregation leading to neurodegenerative diseases [43]. Hence, inhibition of tyrosinase could disrupt the progression of a neurodegenerative disorder by reducing neuromelanin synthesis and preventing oxidative stress, suggesting its potential neuroprotection effect.

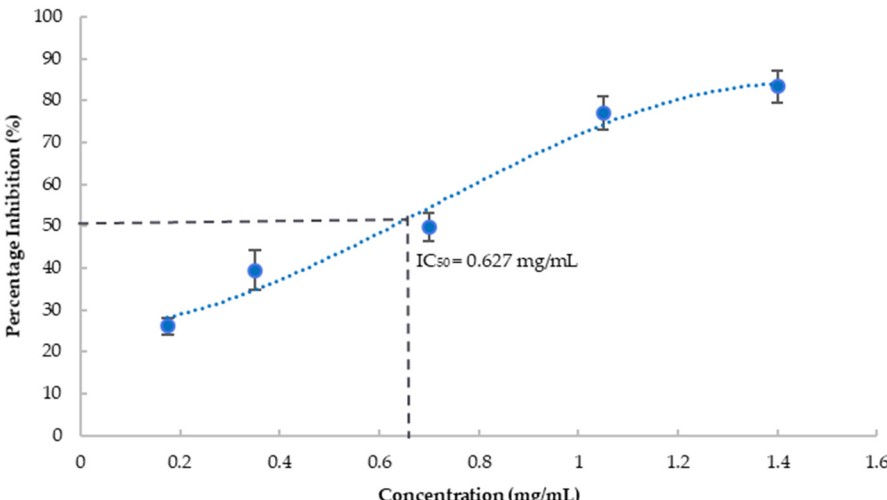

**Figure 2.** Percentage inhibition of tyrosinase by blueberry extract at various concentrations. Values are presented as the mean ± SD of *n* = 6.

- COX-2 inhibition activity:

To investigate the anti-inflammatory effect of blueberries, we evaluated the effect of the blueberry extract on COX-2, a critical enzyme in the inflammation cascade. Most used COX-2 inhibition assays are based on the colorimetric method utilizing the peroxidase activity of COX. Due to the antioxidant property of blueberry extract, it interferes with the widely used fluorometric method for the determination of COX-2 activity (data not shown). Therefore, we chose the competitive ELISA method to monitor the concentrations of prostaglandins formed as a result of the activity of COX-2. Our results showed that blueberry extract strongly inhibits COX-2 activity in a concentration-dependent manner (Figure 3), and the $IC_{50}$ was estimated as 18 μg/mL explicated from the dose-dependent inhibition curve. This indicated that blueberry extract possesses strong anti-inflammatory properties. Inflammation plays a key role in many age-related diseases, including dementia. COX-2 and PGs have pivotal roles in neuroinflammation and stimulate the production of proinflammation cytokines [44]. COX-2 and PGs also mediate apoptosis, autophagy, and loss of synaptic plasticity, leading to neurotoxicity, and AD pathogenesis [21]. Blueberries have been shown to downregulate COX-2 in ovarian cancer SKOV3 cells [45]. To the best of our knowledge, we are the first to report that blueberry can directly inhibit the enzymatic activity of COX-2, with the $IC_{50}$ as low as 18 μg/mL (equivalent to 12 mg/mL fresh berry). As a comparison, the $IC_{50}$ of celecoxib, a strong selective COX-2 inhibitor, is reported as 0.097 μg/mL [46]. While weaker than celecoxib, being food, blueberries can be consumed

in larger quantities. The anti-inflammatory effect of blueberries could directly contribute to their neuronal protection effects.

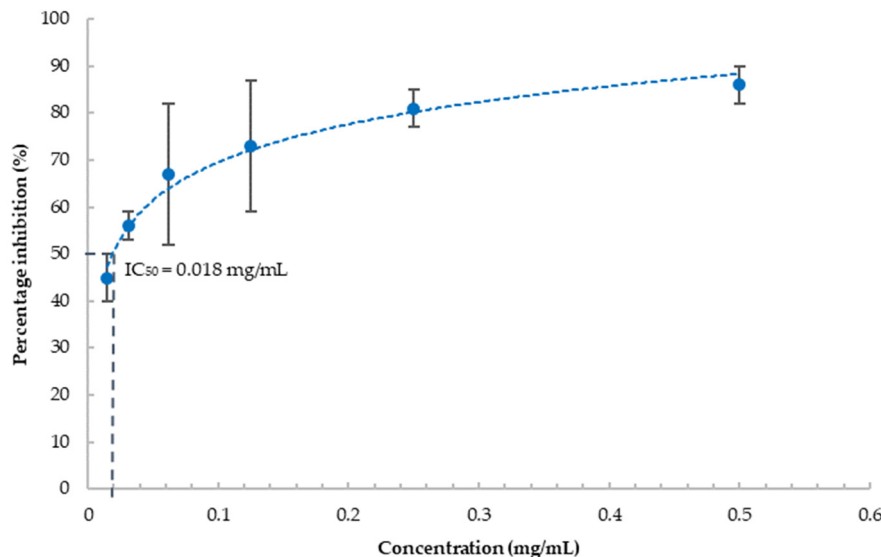

**Figure 3.** Percentage inhibition of COX-2 by blueberry extract at various concentrations. Values are presented as the mean $\pm$ SD of $n = 3$.

### 3.4. Effect of Blueberries on the Synthesis of AChE

To study the effects of blueberry extract at the cell level, the human neuroblastoma cell line M17 was used to study the effect of blueberries on the biosynthesis of AChE. Cell viability was determined using the MTT assay and it was observed that blueberry extract does not affect cell viability below the concentration of 0.06 mg/mL (Figure 4).

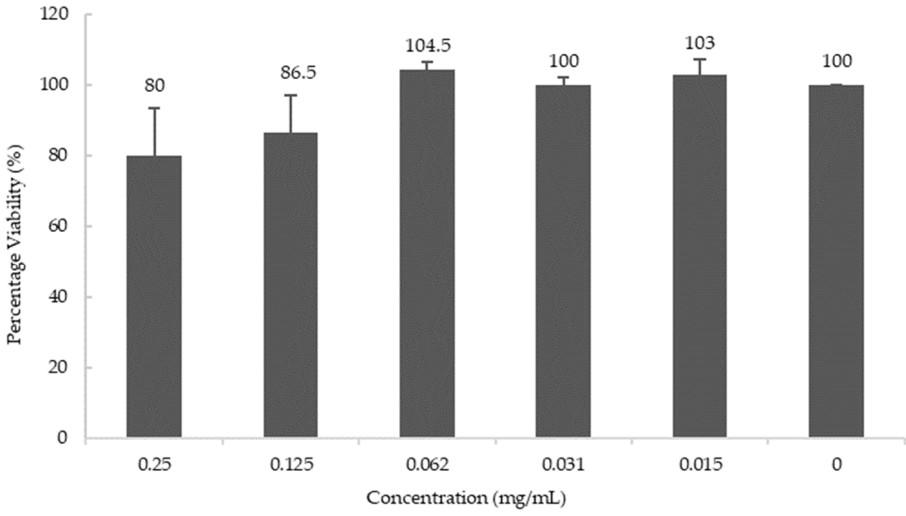

**Figure 4.** Cell Viability at various concentrations of blueberry extract. Values are presented as the mean $\pm$ SD of $n = 6$.

The morphology of untreated cells under the microscope looked round with extended neurites and healthy whereas the treated cells showed considerable shrinkage due to stress when a higher concentration of the blueberry extract was used after 18 h of treatment (Figure 5). At the concentration of 0.06 mg/mL and 0.03 mg/mL of blueberry extract, the morphology of cells is similar to untreated cells (Figure 5), suggesting that below these concentrations of blueberry extract cells are healthy. Therefore, the above two concentrations were employed to study blueberry effects on the biosynthesis of AChE.

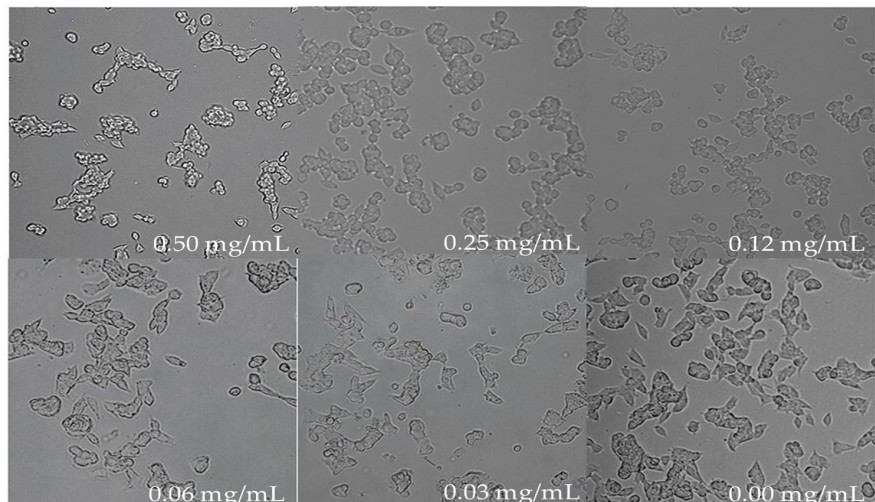

**Figure 5.** Cell morphology post 18 h of treatment at various concentrations of blueberry extract (magnification 10× and resolution 12).

AChE quantitation from cell lysates was performed using the ELISA method. Two methods were used to normalize the results; one is based on the total protein concentration determined by the Bradford method; the other is based on the cell pellet mass. Both methods showed a significant reduction of AChE by blueberry extract compared to untreated cells at both concentrations of blueberry extract ($p = 0.014$ for 0.03 mg/mL and $p = 0.0045$ for 0.06 mg/mL). As shown in Figure 6, the reduction of AChE level of cells treated with 0.06 mg/mL and 0.03 mg/mL of blueberry extracts is 44% and 34%, respectively, based on the normalization of total protein centration, and 46% and 41% based on pellet weight. There are no significant differences between the normalization methods ($p = 0.2938$). While there is a lower trend in reduction with the lower concentration of blueberry extract (0.03 mg/mL vs. 0.06 mg/mL), the difference is not reaching significance ($p = 0.09$). The results here demonstrated that blueberry extract inhibited the synthesis of AChE at the cellular level at as low as 0.03 mg/mL concentration (19 mg/mL in terms of fresh berries). Our results demonstrated that blueberry extract apart from directly inhibiting the enzymatic activity of AChE/BChE also reduces the synthesis of AChE in the ex vivo neuronal cell model. The animal study showed that blueberry extract reduces the brain AChE activity and improves the cognition function of mice [47]. Our data suggested that the reduced AChE activity could be the result of the combined inhibition of the enzymatic activity of AChE and reduced synthesis of AChE by neurons.

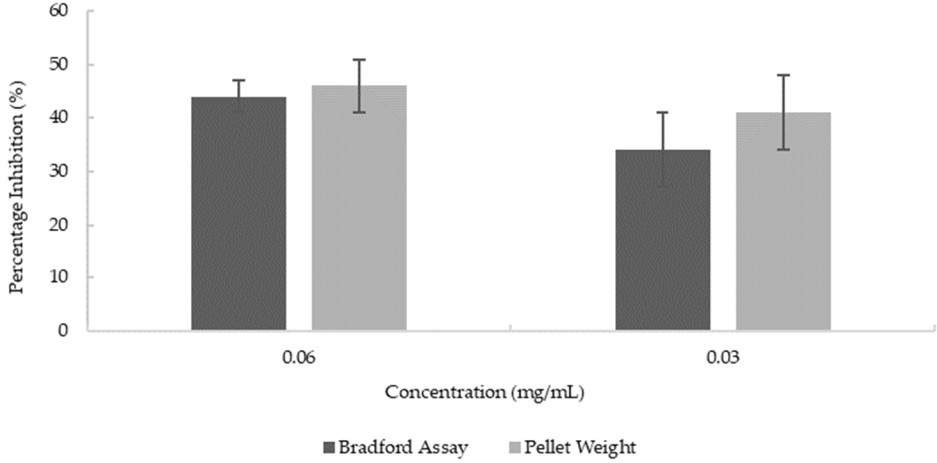

**Figure 6.** Reduction in the synthesis of AChE at various concentrations of blueberry extract. Values are presented as the mean ± SD; $n = 4$ for the pellet weight method and $n = 6$ for the Bradford assay.

The cholinergic system plays an important role in many forms of dementia, including AD. The deficit in cholinergic transmission could affect cognition and behavior, leading to dementia [48]. Inhibition of AChE/BChE could counter the cholinergic deficit and improve cognition [14,49]. In addition to directly contributing to the cholinergic deficit, AChE is at an elevated level in amyloid plaque and neurofibrillary tangles regions, suggesting AChE can modulate the Aβ precursor protein processing and production, particularly in a degenerative cycle that enhances Aβ precursor protein processing, leading to the formation of amyloid plaques [10]. Our results on the dual inhibition of AChE/BChE and reduced AChE biosynthesis by blueberries (at low mg/mL fresh berry equivalent) provide the basis that blueberries play important roles in improving cholinergic signaling, and potentially modulate the Aβ precursor protein processing and production.

### 3.5. Anti-Amyloidogenic Property

Amyloid aggregation is associated with several degenerative diseases affecting the brain or peripheral tissues. Amyloid aggregation in the brain is one of the main hypotheses for AD [8,50]. To examine the effects of blueberries on amyloid fibril formation, we used HEWL as a model. It is well known that HEWL solution forms amyloid fibrils in acidic conditions with high temperatures [35,51]. Thioflavin T (ThT) is among the widely used fluorescence probes for selectively staining and identifying amyloid fibrils both in vivo and in vitro by strong fluorescence at 490 nm upon binding to amyloid fibrils. However, most polyphenols absorb the light around 490 nm and significantly quench ThT's fluorescence [52,53]. Our own data also showed blueberries quench ThT fluorescence at 490 nm dramatically (data not shown). Therefore, using ThT to monitor the inhibition of the amyloid aggregation by polyphenols will generate bias and false positive results.

Congo red is widely used as a stain to amyloid fibrils and the binding to amyloid fibrils leads the CR absorption to a red shift with a unique shoulder peak near 540 nm [54]. As shown in Figure 7A, following agitation (170 rpm) and elevated temperature (70 °C) at low pH (pH 2.0 glycine buffer), the aged HEWL induced the absorbance of CR to a longer wavelength with the appearance of the secondary peak at 536 nm indicated the formation of amyloid fibril. Hence, the secondary peak of CR following binding to amyloid fibril at 536 nm was used to detect the amyloid formation.

Figure 7B,C show the CR spectra with HEWL alone and that with the blueberry-treated sample in glycine buffer, pH 2.0. It clearly showed that HEWL alone formed amyloid fibrils after 24 h (day 1) (Figure 7B), while the blueberry-treated sample formed amyloid fibrils after 48 h (day 2) (Figure 7C). Since fresh HEWL without amyloid fibril does not show the shoulder peak at 536 nm (Figure 7), the ratio of the absorbance at 536 nm and 509 nm can be used as the marker for amyloid fibrils. A ratio approaching the value of 1 indicates more amyloid fibrils formed in the sample. As shown in Figure 8, on day 1, after 24 h, there is a significant difference between the ratio of the HEWL in glycine in comparison to the HEWL treated with blueberries, suggesting that blueberries prevent the HEWL from forming amyloid fibrils during the first 24 h.

To further confirm the findings from CR binding, we used Nile red (NR) to probe the effects of blueberry on the amyloid genesis and surface hydrophobicity of HEWL. NR is a hydrophobic dye whose fluorescence emission wavelength and intensity vary depending on the polarity of the environment [55,56]. Its fluorescence is quenched in an aqueous solution. In the hydrophobic environment, the wavelength of the NR fluorescence moves to a shorter wavelength (blue shift) with the increase of the fluorescence intensity [57]. NR's fluorescence is independent of pH. During the amyloid aggregation, the overall structure of protein changes, and the hydrophobic regions are exposed. Therefore, NR becomes a versatile extrinsic probe for amyloid detection. The longer emission wavelength of NR (above 600 nm) can also reduce the interferences of polyphenols and other small molecules, making it possible to use NR screening anti-fibril drugs against amyloid diseases [57]. The aged HEWL sample at high temperature and low pH showed an increase in the fluorescence of NR with the blueshift of the maximum emission wavelength (from 652 nm to 625 nm) as

depicted in Figure 9A. This observation is in agreement with the observations from the CR binding assay, confirming the formation of amyloid aggregation in aged samples. Similar to the findings from CR, the treatment of blueberry prevents amyloid fibril formation in the first 24 h and the blue shift of emission wavelength starts on day 2 (Figure 9B,C).

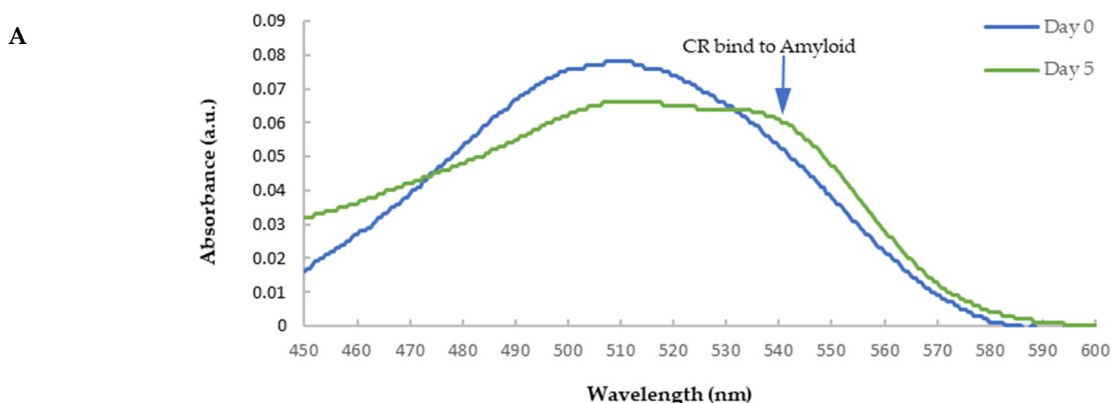

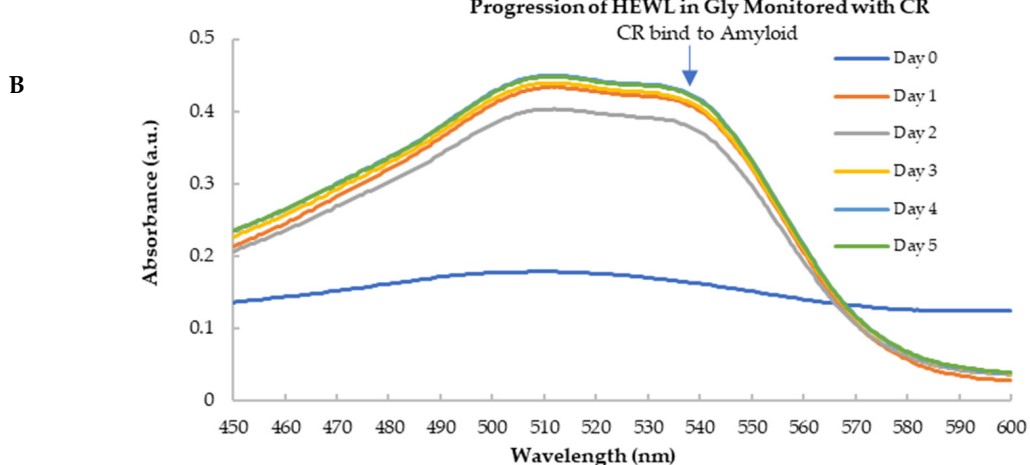

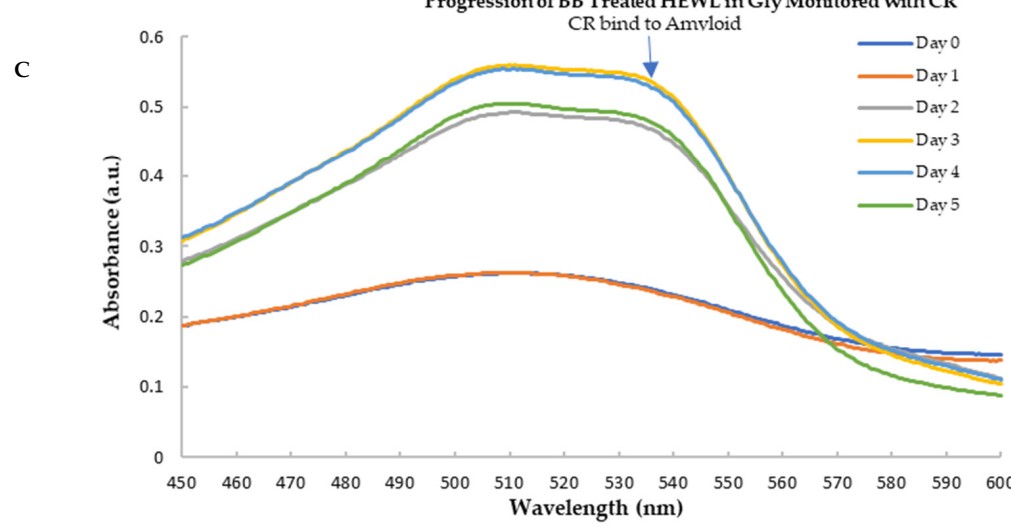

**Figure 7.** Spectra of CR with HEWL and blueberry-treated HEWL. (**A**) The spectra of CR with HEWL (normalized against day 0). The amyloid fibril induced a 536 nm peak in the day 5′s sample. (**B**) The spectra of CR with HEWL alone during 5 days; (**C**) The spectra of CR with blueberry-treated HEW during 5 days.

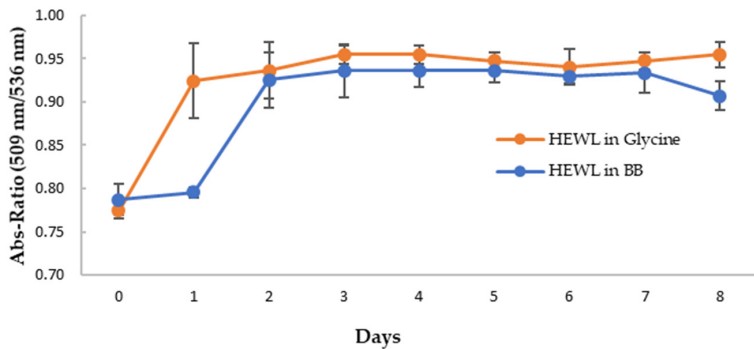

**Figure 8.** The average ratio of absorbance at 509 nm and 536 nm of CR with HEWL alone (orange) and blueberry-treated HEWL (blue). The error bars represent the SD of three independent runs.

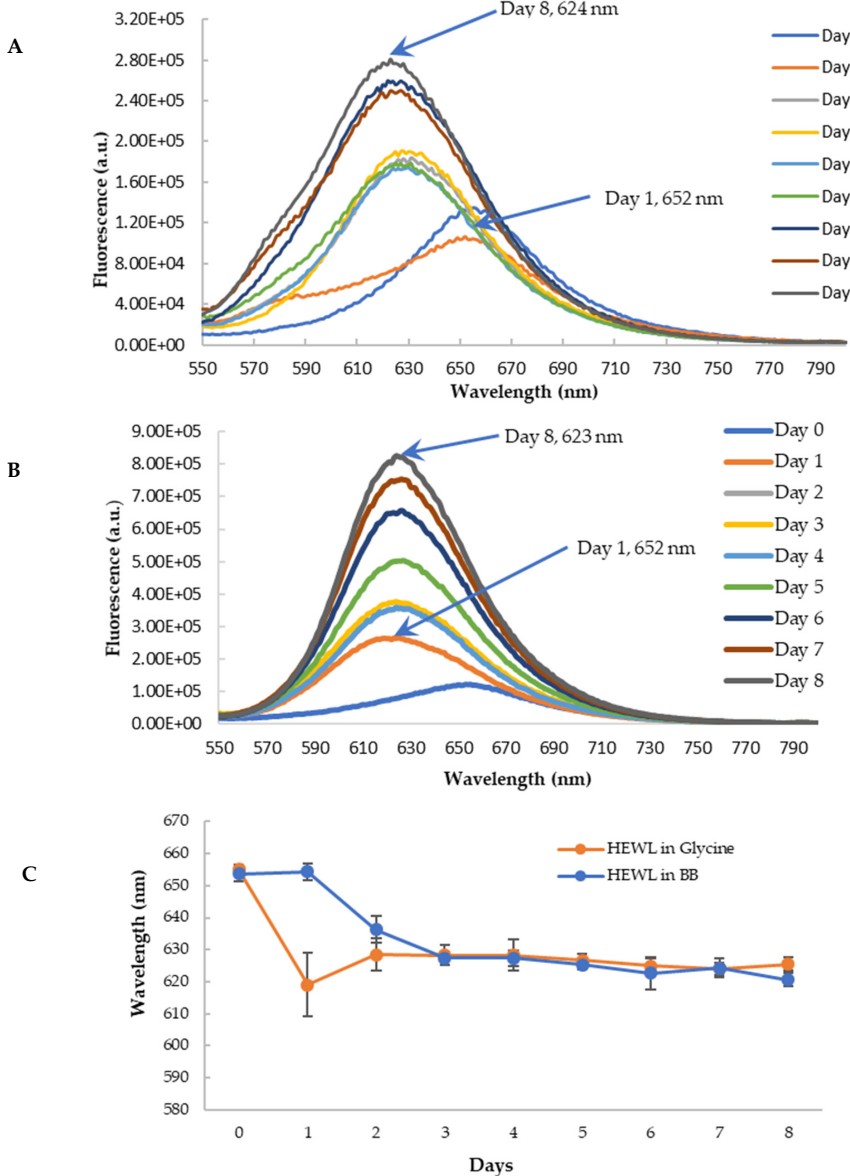

**Figure 9.** Fluorescence emission spectra of Nile Red with HEWL in glycine buffer with excitation at 530 nm. (**A**): HEWL alone; (**B**): Blueberry-treated HEWL; (**C**): The maximum emission wavelength of HEWL alone (orange) and blueberry-treated HEWL (blue). The error bars represent the SD with three independent runs.

In addition to the blue shift of the emission wavelength, the intensity of the fluorescence of NR was also increased upon binding to amyloid fibrils (Figure 9A), suggesting NR binding to the hydrophobic surface of HEWL during the formation of amyloid fibrils. The extent of increase for blueberry-treated HEWL, however, is less (Figure 9B). The fluorescence intensity of NR is dependent on their exposure to the hydrophobic environment. Therefore, the intensity of fluorescence of NR upon binding to the amyloid fibrils provides a means to quantify the extent of the hydrophobic core exposure during amyloid fibril formation. To remove the interferences from blueberries, and more accurately quantify the reduction in amyloid aggregation by blueberries, the fluorescence spectra of HEWL samples were corrected by spiking the samples with the blueberry extract at the same concentration as the treated samples only during the recording of the fluorescence spectra (but not during the aging process). The maximum emission wavelength of NR was not affected by spiking the control samples with the blueberries. Based on the corrected emission intensity (Figure 10), the fluorescence of the NR binding to the amyloid fibrils in the blueberry-treated sample is less than that of HEWL alone, suggesting that blueberries reduce the levels of amyloid fibrils under amyloidogenic conditions. The reduction is clearer on days 6 and 7. The fluorescence of NR-HEWL in glycine decreases on day 8 (Figure 10), which is possibly due to the more aggregations formed, and therefore, fewer hydrophobic regions of HEWL exposed to NR.

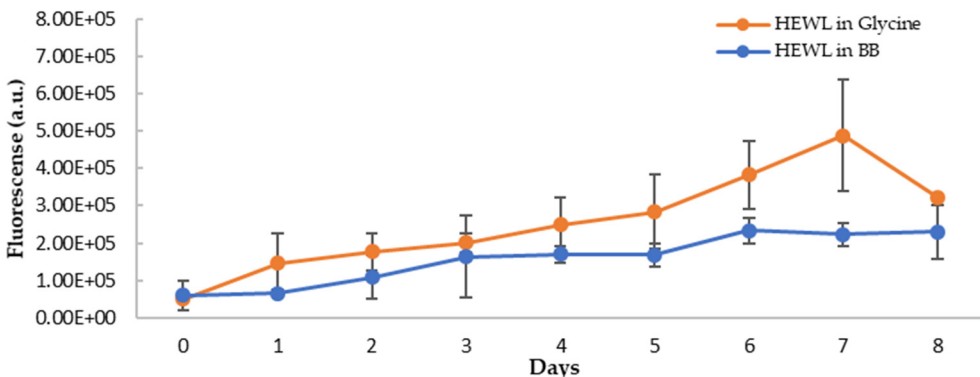

**Figure 10.** Fluorescence intensity of HEWL alone (orange) and blueberry-treated HEWL (blue) samples post-correction (emission at 623 nm, excitation at 530 nm). The error bars represent SD with three independent runs.

The visual inspection of the aged HEWL treatment samples showed more turbidity than the fresh HEWL, suggesting a correlation between the turbidity and amounts of aggregates in the samples. The turbidity of HEWL alone is more than the blueberry-treated sample by visual examination. To quantify the turbidity, we monitored the light scattering using the fluorometer by using the same wavelength (800 nm) for excitation and emission. The reading at a 90-degree angle from the light source in the fluorometer removes the interferences from the sample absorbance. The long wavelength (800 nm, near IR range) was chosen to further reduce any sample absorbance interferences. The respective blanks were subtracted from samples to correct any background interference. As shown in Figure 11, the turbidity of both samples increases over the eight-day period, however, the HEWL in glycine has a higher degree of turbidity. On day 7 and day 8, the turbidity of the blueberries-treated sample is only about 63% and 60% of HEWL controls, respectively, suggesting a 37–40% decrease in amyloid aggregation. This trend is the same as in NR binding assay, suggesting that blueberries not only delay the amyloid fibril formation by 24 h but also slow down the progression of amyloid fibrils during the aging process. The decrease in light scattering on day 8 is possibly due to the settlement of the aggregation during the assay.

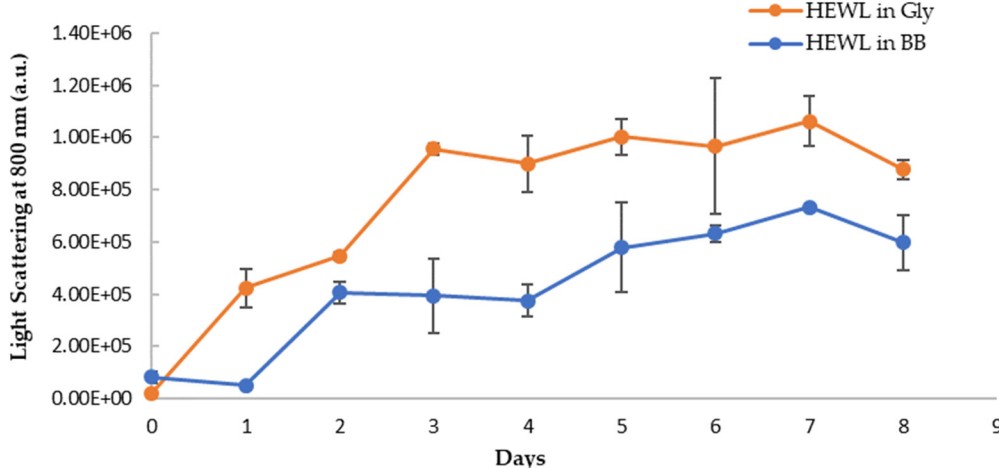

**Figure 11.** Turbidity (emission at 800 nm with excitation at 800 nm) of HEWL alone (orange) and blueberry-treated HEWL (blue) in glycine buffer. The error bars represent SD with three independent runs.

Both CR and NR binding assays are consistent in that blueberries delay amyloid aggregation for 24 h. The NR binding and light scattering also showed that blueberries reduced the amyloid fibril level. These results support that blueberries inhibit the progression of aggregation and demonstrated that as low as 0.06 mg/mL blueberry extract (equivalent to 39 mg/mL fresh berry) can reduce amyloid fibril formation. To our best knowledge, this is the first report that blueberry extract directly inhibits amyloid fibril formation.

Despite different precursors and diversified amino acid sequences of amyloidogenic peptides or proteins, they misfolded into amyloid fibrils by sharing a common core motif characterized by the cross-β-sheet structure, including Aβ aggregation [57,58]. While we used the HEWL as a model for amyloid formation, our results implicate that blueberry may also inhibit the Aβ aggregation and protect cytotoxicity of Aβ aggregation to neurons. The reduced Aβ aggregation may further reduce AChE level and enhance cholinergic signaling [11].

## 4. Conclusions

Our data from this study showed that blueberry substantially inhibits the enzymatic activity of AChE, BChE, tyrosinase, and COX-2 in vitro, and reduces the synthesis of AChE in ex vivo cellular model at μg/mL level (mg/mL level in terms of fresh berries). Through inhibition of these enzymes, blueberries could enhance cholinergic signaling, and further modulate the oxidation stress and APP processing, reduce neuroinflammation and confer neurocognition. Furthermore, through the HEWL model, polyphenol-rich blueberries have shown the anti-amyloidogenic property by reducing the progression of the amyloid fibril formation, which could lead to reducing the toxicity of amyloid to neurons. These results provide more molecular mechanisms of the neuroprotective effects of blueberries beyond their antioxidant activities. AD (and other neurodegenerative diseases) is a complex disease and not caused by any single factor. Thus, targeting just one pathway may not be adequate to treat or prevent AD and other neurogenerative diseases. Blueberries, as demonstrated in this study, affect multiple enzymes and pathways. Therefore, consuming blueberries, as part of a healthy lifestyle, may play a pivotal role in the prevention and/or delaying of the progression of neurodegenerative diseases. We chose blueberries from the marketplace in this study, and therefore, our findings are more relevant to the health benefits consumers have with the consumption of commercially available blueberries and support blueberries being nutraceutical to protect neurons and improve cognitive function.

**Author Contributions:** S.C. (Shuowei Cai) and P.S. designed the research; P.S. and S.C. (Sophia Costa) conducted the research; S.C. (Shuowei Cai), P.S. and S.C. (Sophia Costa) wrote the paper. S.C. (Shuowei Cai) supervised the project. All authors have read and agreed to the published version of the manuscript.

**Funding:** This research received no external funding.

**Institutional Review Board Statement:** Not applicable.

**Informed Consent Statement:** Not applicable.

**Data Availability Statement:** The data presented in this study are available on request from the corresponding author.

**Acknowledgments:** Pari Samani and Sophia Costa were supported through a Thesis Support Grant from the College of Arts and Sciences at the University of Massachusetts Dartmouth. Sophia Costa was also supported by a grant from the Office of Undergraduate Research (OUR) at the University of Massachusetts Dartmouth.

**Conflicts of Interest:** The authors have no conflict of interest to report.

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
