# Peer review of "Neuroprotective Effects of Blueberries through Inhibition on Cholinesterase, Tyrosinase, Cyclooxygenase-2, and Amyloidogenesis"

_nutraceuticals, doi:10.3390/nutraceuticals3010004_

Round 1

Reviewer 1 Report

1. overall scientific quality is ok

2. introduction mainly concentrate on bluberry and AD, while in title other actvity also mention, improve your introduction, also add rational of the study.

3. kindly check this reference 

Effect of a polyphenol-rich wild blueberry extract on cognitive performance of mice, brain antioxidant markers and acetylcholinesterase activity. 

Behavioural Brain Research

Volume 198, Issue 2, 17 March 2009, Pages 352-358

4. Conclusion need refining

Author Response

We’d like to sincerely thank the reviewer for their thoughtful comments. Now we revised our manuscript accordingly.

  1. overall scientific quality is ok

We’d like to thank the reviewer for the comments.

  1. introduction mainly concentrate on bluberry and AD, while in title other actvity also mention, improve your introduction, also add rational of the study.

We added the introductions of cholinesterase, neuroinflammation, and COX-2, the interconnected hypotheses of oxidation stress, neuroinflammation, cholinergic signaling, and beta-amyloid to the introduction session (lines 55-80), and added the rationale and aims of the study (lines 98-104).

  1. kindly check this reference

Effect of a polyphenol-rich wild blueberry extract on cognitive performance of mice, brain antioxidant markers and acetylcholinesterase activity.

Behavioural Brain Research

Volume 198, Issue 2, 17 March 2009, Pages 352-358

Thank you for providing the reference, and we now added this reference to the results and discussion section (lines 404-497).

  1. Conclusion need refining

Thank you for the reviewer’s suggestion. We refined the conclusion section (line 654-677).

Reviewer 2 Report

Introduction

The authors should clearly state the main aim of their work.

Authors should check carefully the following papers and make use of, to enrich their introduction and discussion.

https://doi.org/10.3390/nu14081619

https://doi.org/10.3390/plants11040568

https://doi.org/10.3390/foods9060690

https://doi.org/10.3390/molecules26185582

Materials and methods

In a bid to develop nutraceutical, why did the authors include acetone and methanol in the extraction solvent? 40% of acetone and 40% of methanol will lead to toxicity and will disqualify the good assays and interesting results obtained. Methanol and acetone will hardly be evaporated efficiently to dryness using a rota vapor as indicated.

Line 110: correct rotor evaporator.

Addition of monoamine oxidase inhibition and BChE inhibition will complement the work.

Add the microscope to the materials

Results and discussion

The yield is very low

The half-maximal inhibitory concentration (IC50) is wrong because the maximal is not necessarily 100%. Line 301

Author Response

We’d like to sincerely thank the reviewer for their thoughtful comments. Now we revised our manuscript accordingly.

Introduction

The authors should clearly state the main aim of their work.

Reply: Thank you for the suggestions. We are now adding the rationale and aims for this study.

Authors should check carefully the following papers and make use of, to enrich their introduction and discussion.

https://doi.org/10.3390/nu14081619

https://doi.org/10.3390/plants11040568

https://doi.org/10.3390/foods9060690

https://doi.org/10.3390/molecules26185582

 Reply: Thank you for providing the references. We added those references in the introduction section and results and discussion section.

Materials and methods

In a bid to develop nutraceutical, why did the authors include acetone and methanol in the extraction solvent? 40% of acetone and 40% of methanol will lead to toxicity and will disqualify the good assays and interesting results obtained. Methanol and acetone will hardly be evaporated efficiently to dryness using a rota vapor as indicated.

Reply: Thank you for the valuable comments. In addition to rotary evaporation, we also did lyophilization. Despite the rotary evaporation and lyophilization, there is still the possibility of having toxic residual solvents. Indeed, we have explored other solvents, and find the non-toxic solvent ethanol can effectively extract polar compounds from blueberries and we will publish those results soon. We added this discussion in lines 232-327.

Line 110: correct rotor evaporator.

Reply: Thank you for pointing out the typo. We corrected this in the Materials and Methods section.

Addition of monoamine oxidase inhibition and BChE inhibition will complement the work.

Reply: Thank you for the valuable suggestions. There is a report that blueberry can inhibit monoamine oxidase, we added this information in the introduction section (line 98-100). We carried out the inhibition assay on BuChE, and reported the results in this report.

Add the microscope to the materials

Reply: Thank you for the suggestion. We added this information to the materials and methods section (lines 131-134).

Results and discussion

The yield is very low

Reply: Thank you for this comment. We did the extraction on the fresh berry, which contains 84$ water, 14.5% carbohydrate, 0.74% protein, and 0.33% fat. There are only less than 0.5% mass of fresh berries as micronutrients (less than 2.5 g of 500 g fresh berries) in fresh berries, and our extract yielded 778 mg polar extract. We added that information in the results and discussion section (lines 317-323).

The half-maximal inhibitory concentration (IC50) is wrong because the maximal is not necessarily 100%. Line 301

Reply: Thank you for the valuable comment. We apologize for the confusion. We now defined IC50 as the absolute 50% inhibition of the enzymatic activity of respective enzymes in method section (lines 183-185).

Reviewer 3 Report

A research paper on the effect of blueberries in its role as neuroprotectors is presented. The objectives are interesting, but the manuscript needs to be improved. Here are some moderate points to attend to:

1.      The inhibition/reduction formulas are basically the same expression. It is suggested to describe it in a general way, if possible. The formula has a different format than the text, homogenize.

2.      All figures that are currently in histogram format should be displayed in scatter point format. If an adjustment was made to IC50, the adjustment line (dotted) and the IC50/DE50 value should be placed, as the case may be.

3.      Make sure that the resolution of all the figures is at least 300dpi. Some of them look blurry. Do not place borders on figures.

4.      Callouts in figures should not exceed the size of the main text.

5.      In the % inhibition/reduction graphs, the average percentage was placed above the bar. I don't think it is necessary, but if it is left then the value of the average and its deviation should be noted. Likewise, the average percentage and its SD value should be mentioned in the main text (currently there is only the average value, without dispersion values).

6.      Very important, the type of statistics used is not explained. If it is only descriptive, without making comparisons, the conclusions could be less categorical. If comparisons have been made using statistical tests (which is the most appropriate), these should be described in methods (type of comparisons, tests used to compare, tests to verify normality or homoscedasticity of the data, sample size calculation, etc.)

a.      For example, in line 380 it says that “Our results demonstrated that blueberry extract apart from directly inhibiting the enzymatic activity of AChE also reduces the synthesis of AChE in the ex-vivo neuronal cell model.” However, there are no strict comparisons (statistical comparisons) that support the “demonstrated” statement. Please, include statistical formality or change the categorical sentences for suggestions according to the observations.

7.      An interesting aspect is mentioned from line 365 “The morphology of untreated cells under the microscope looked round with extended neurites and healthy whereas the treated cells showed considerable shrinkage due to stress when a higher concentration of the blueberry extract was used...", which seems to me to be relevant information and could be in the manuscript. I strongly suggest showing representative images (if possible, also quantifying some morphological parameter) of the mentioned cells.

8.      Figure 7 and 8 can be merged into one, since figure 8 is the quantification of the two graphs of figure 7.

9.      Use scientific notation on figure axes 9, 10, and 11 (in arbitrary units), or normalize. In subsection A of figure 9, the value mentioned on day 1 is incomplete.

1.   the title can be fixed. Something like Neuroprotective effect of blueberries through Inhibition on Ace-2 tylcholinesterase, Tyrosinase, Cyclooxygenase-2, and Amyloi-3 dogenesis is suggested

Author Response

We sincerely thank the reviewer for the insightful comments. We revised our manuscript accordingly.

A research paper on the effect of blueberries in its role as neuroprotectors is presented. The objectives are interesting, but the manuscript needs to be improved. Here are some moderate points to attend to:

  1. The inhibition/reduction formulas are basically the same expression. It is suggested to describe it in a general way, if possible. The formula has a different format than the text, homogenize.

Reply: Thank you for the suggestion. We consolidate the inhibition calculation in equation 1, and homogenize the formula with the same format as the main text.

  1. All figures that are currently in histogram format should be displayed in scatter point format. If an adjustment was made to IC50, the adjustment line (dotted) and the IC50/DE50 value should be placed, as the case may be.

Reply: Thank you for the suggestion, We changed the figures of IC-50 from histogram to scatter plots, with the non-linear regression lines for the calculation, and draw the dotted lines to indicate the IC50 values.

  1. Make sure that the resolution of all the figures is at least 300dpi. Some of them look blurry. Do not place borders on figures.

Reply: Thank you for the suggestions, and we apologize for the blurry figures. Now we improve the resolution on all figures.

  1. Callouts in figures should not exceed the size of the main text.

Reply: Thank you for the suggestions, and we apologize for the inconsistency on the callouts in the figures. We now put all the callouts in the figures smaller than the main text font.

  1. In the % inhibition/reduction graphs, the average percentage was placed above the bar. I don't think it is necessary, but if it is left then the value of the average and its deviation should be noted. Likewise, the average percentage and its SD value should be mentioned in the main text (currently there is only the average value, without dispersion values).

Reply: Thank you for the suggestions. We removed the value of the average from the figures.

  1. Very important, the type of statistics used is not explained. If it is only descriptive, without making comparisons, the conclusions could be less categorical. If comparisons have been made using statistical tests (which is the most appropriate), these should be described in methods (type of comparisons, tests used to compare, tests to verify normality or homoscedasticity of the data, sample size calculation, etc.)

Reply: Thank you for the valuable comment. We added the statistical program used in the method part (line 265),

  1. For example, in line 380 it says that “Our results demonstrated that blueberry extract apart from directly inhibiting the enzymatic activity of AChE also reduces the synthesis of AChE in the ex-vivo neuronal cell model.” However, there are no strict comparisons (statistical comparisons) that support the “demonstrated” statement. Please, include statistical formality or change the categorical sentences for suggestions according to the observations.

Reply: We appreciate the reviewer’s valuable comments and add the discussion on the statistical analysis in the results and discussion section (lines 481-497).

  1. An interesting aspect is mentioned from line 365 “The morphology of untreated cells under the microscope looked round with extended neurites and healthy whereas the treated cells showed considerable shrinkage due to stress when a higher concentration of the blueberry extract was used...", which seems to me to be relevant information and could be in the manuscript. I strongly suggest showing representative images (if possible, also quantifying some morphological parameter) of the mentioned cells.

Reply: Thank you for this valuable suggestion. We added the cell morphology figure in the manuscript (Fig. 5).

  1. Figure 7 and 8 can be merged into one, since figure 8 is the quantification of the two graphs of figure 7.

Thank you for the valuable suggestion. We combined Fig 7 and Fig 8 into one figure.

  1. Use scientific notation on figure axes 9, 10, and 11 (in arbitrary units), or normalize. In subsection A of figure 9, the value mentioned on day 1 is incomplete.

Reply: Thank you for the suggestions. The absorbance is unitless and for the fluorescence and light scattering, the unit is arbitray unit (AU).

  1. the title can be fixed. Something like Neuroprotective effect of blueberries through Inhibition on Acetylcholinesterase, Tyrosinase, Cyclooxygenase-2, and Amyloidogenesis is suggested

Thank you for the valuable suggestion. We revised the title as the “Neuroprotective Effect of Blueberries…..”

Round 2

Reviewer 2 Report

The authors have carefully implemented the corrections and the paper is good in the present form.